# Automatic segmentation of the pulmonary lobes with a 3D u-net and optimized loss function

**Bianca Lassen-Schmidt**[1]      BIANCA.LASSEN@MEVIS.FRAUNHOFER.DE
**Alessa Hering**[1]      ALESSA.HERING@MEVIS.FRAUNHOFER.DE
**Stefan Krass**[1]      STEFAN.KRASS@MEVIS.FRAUNHOFER.DE
**Hans Meine**[1]      HANS.MEINE@MEVIS.FRAUNHOFER.DE
[1] *Fraunhofer MEVIS, Bremen, Germany*

## Abstract

Fully-automatic lung lobe segmentation is challenging due to anatomical variations, pathologies, and incomplete fissures. We trained a 3D u-net for pulmonary lobe segmentation on 49 mainly publically available datasets and introduced a weighted Dice loss function to emphasize the lobar boundaries. To validate the performance of the proposed method we compared the results to two other methods. The new loss function improved the mean distance to 1.46 mm (compared to 2.08 mm for simple loss function without weighting).

**Keywords:** pulmonary lobes, lung lobes, segmentation, deep learning, CNN, 3D U-net

## 1. Introduction

The human lungs are subdived into five lobes with separated supply branches for both vessels and airways. Usually, a visceral pleura called the pulmonary fissure can be found between adjacent lobes. Accurate segmentation of the five pulmonary lobes is important for diagnosis, treatment planning and monitoring for lung diseases such as COPD and fibrosis. In the last two decades several automatic approaches were developed (Doel et al., 2015) each has different drawbacks and limitations. A few recently published approached are based on deep learning (Gerard and Reinhardt, 2019; Wang et al., 2019; Tang et al., 2019; Park et al., 2019; Lee et al., 2019; Imran et al., 2018; George et al., 2017). But none of them includes explicit knowledge from pulmonary fissures as weighting into the loss function. In this paper, we propose a fully automatic lobe segmentation method based on a 3D u-net with an optimized loss function focussing the lobar boundaries. We compare the new method with a proven fully automatic segmentation approach (Lassen et al., 2013) that won the LOLA11 challenge (LOLA11) at publication time.

## 2. Method & Data

Pulmonary fissures are the most accurate feature for lobe segmentation but these are often incomplete and there can be accessory fissures in the lungs. We train a 3D u-net to rely on visible fissures and simultaneously learn that fissures are often incomplete.

*Data:* We use 70 lung CT scans, randomly subdivided into 49 training, 7 validation and 14 testing data sets. 50 datasets are taken from LIDC/IDRI (Armato et al., 2011) with public available reference segmentations by Tang et al. (Tang et al., 2019) and 20 datasets

are from the University Medical Center Utrecht that are also used in (Lassen et al., 2013). Reference segmentations for the 20 datasets are generated with an automatic segmentation and interactive correction.

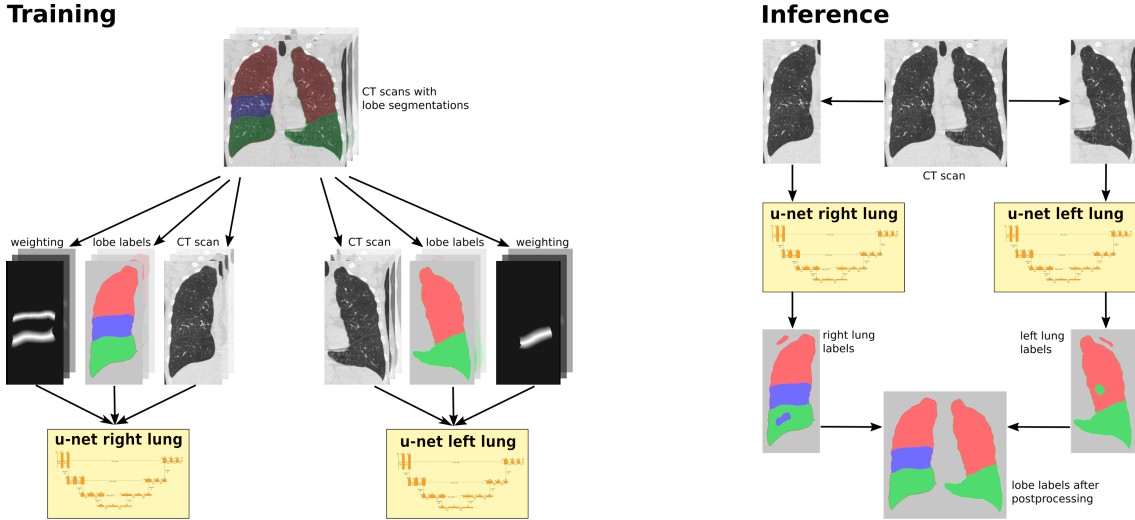

Figure 1: Training and segmentation process of the proposed method.

*Training:* We train two separate 3D u-nets (see Figure 1), one for the left lung with 3 labels (background, upper lobe, lower lobe) and one for the right lung with 4 labels (background, upper lobe, middle lobe, lower lobe). We use 4 resolution levels, a filter size of $3^3$ for the convolutions, PReLu activation, batch normalization and dropout.

The most promising features for lung lobe segmentation can be found at the lobar boundaries. We want our networks to learn two facts: 1. pulmonary fissures are at the lobar boundaries, but 2. there might be no fissures at the lobar boundaries (incomplete fissures). Therefore we introduce a weighted Dice loss in the following manner: We calculate an Euclidean distance transformation (EDT) from the lobar boundaries of the reference segmentation. Next, we invert the distance *dist* within a radius of 10 mm by setting the values to *10 - dist* and set the remaining part of the lungs to a value of 1. This results in a weighting image $w$ with a value of 10 at the lobar boundaries descending to value 1 within 10 mm radius. We use the EDT because the reference segmentation might not be exact on the lobar boundaries and we do not want to miss information from visible fissures. Thus, we use the following loss function: $WeightedDiceLoss = 1 - \frac{2\sum_i^N pred_i ref_i w_i}{\sum_i^N pred_i^2 w_i + \sum_i^N ref_i^2 w_i}$

As a preprocessing step the input images are resampled to 1.5 $mm^3$. The 49 training datasets are subdivided into patches of size $60^3$ voxels with 44 voxels padding on all six sides. We start the training with a learning rate of 0.005, a batchsize of 2 and stop after 70000 iterations.

*Inference:* For segmentation, we first resample a CT scan to 1.5 $mm^3$. Next, we apply our trained networks for the left and the right lung lobes and upsample the image back to the original resolution. We propose the following postprocessing to deal with remaining misclassifications regions (see Figure 1): We perform a connected component analysis and

keep the largest two (three) components in the left (right) lung. Finally, we fill the holes with a Voronoi division and use a given lung mask to delete all objects outside the lungs.

## 3. Evaluation & Results

We applied the described segmentation pipeline to the 14 testing datasets which were not used for training and validation. Segmentation including postprocessing takes less than 6 seconds for a case. We compared our method to two other approaches: 1. a non-deep-learning-based automatic method (Lassen et al., 2013) 2. the same u-net as proposed but without weighting. The mean distance from the visible fissure improved to 1.46 mm (without weighting: 2.08 mm). See Figure 2 for plots and Figure 3 for screenshots.

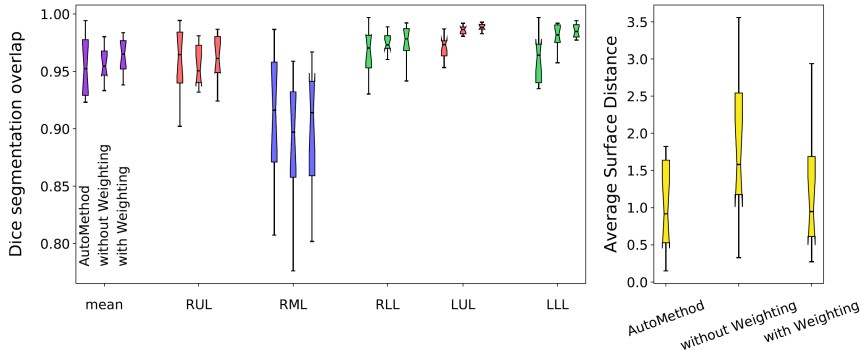

Figure 2: Proposed method in comparison to a non-DL-based automatic method and a 3D u-net without weighting a) Dice coefficient b) mean distance to visible fissures

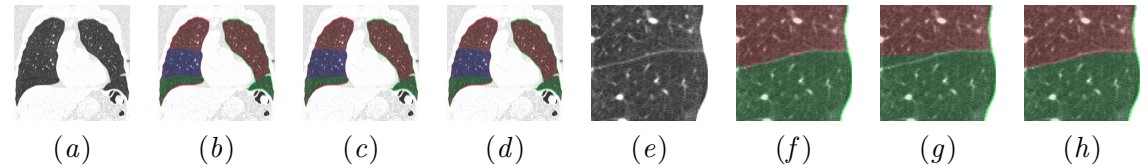

Figure 3: Segmentation results for two cases. a+e) CT scan, b+f) non-DL-based automatic method, c+g) 3D u-net without weighting, d+h) 3D u-net with weighting

## 4. Discussion

We trained a 3D u-net for a lung lobe segmentation task and showed that emphasizing the lobar boundaries in the loss function improved the segmentation results (see Figure 2 and 3). The segmentation quality is comparable to the method proposed in (Lassen et al., 2013) and even slightly better for the left lobes. This study was performed on a small amount of data. In future work, we plan to train with the same architecture on a much larger database including a wide range of pathologies and performing an extensive evaluation with participation in the LOLA11 (LOLA11) challenge.

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
