# OpenReview forum: "Automatic segmentation of the pulmonary lobes with a 3D u-net and optimized loss function"
_MIDL.io/2020/Conference — MIDL 2020_

### Official Review · AnonReviewer2 · 2020-03-10
**The authors propose a CT lobar segmentation method using a deep learning framework. The well-known U-net architecture was utilized with a weighted loss  function that emphasizes the fissure. There is very little if any novelty in the method considering similar deep learning based lobe segmentations have already been proposed with better results. That being said, this work seems satisfactory for a conference but not suitable for journal publication even with more training/evaluation data.**

**Rating:** 3
**Confidence:** 5

**Review:**

In the introduction the authors state that none of the existing deep learning approaches to lobe segmentation use explicit knowledge from pulmonary fissures. This is not true, the Gerard and Reinhardt 2019 reference uses pulmonary fissures as an input channel. This method is currently the leader in the LOLA11 challenge which should also be mentioned.

It seems the proposed method requires a lung segmentation as a precursor which distinguishes left and right lungs (for cropping input). This needs to be explicitly mentioned. It should also be explained in the methods how this was obtained in this work.

During training patches of size 60 are used, however, it is unclear what is done during inference. Are the same patch sizes used? Are non-overlapping patches used? If not, how are patch results merged?

Figure 2 shows the "mean distance to visible fissure". It is unclear how this is calculated. For this to be calculated the evaluation data ground truth would need to include annotations of just visible fissures, however, based on the description it seems only complete lobe segmentations are available, i.e., all extracted fissures would be extrapolated and include both visible and non-visible fissures.

---

### Official Review · AnonReviewer4 · 2020-03-12
**Weighted dice loss based on distance transform to improve fissure segmentations with unconvincing results.**

**Rating:** 2
**Confidence:** 5

**Review:**

Summary:
Modified dice loss with weights computed based on Euclidean distance transform is proposed to improve segmentation of fissures between lung lobes. The weighted dice loss is presented as a novel contribution and is used to train a 3D Unet. Experiments compare performance to a baseline model and with Unet trained without weighted dice loss.

Strengths:
+ Weighting based on Euclidean distance transform is a useful idea.

Weakness:
- Presenting the weighting strategy as a novel idea is a bit of a stretch. The paper could be strengthened with more thorough validation and discussion of the improvements due to weighting.
- The results are not convincing. While weighting dice loss shows improvement to Unet performance, it is still similar to the baseline method. There is no acknowledgement or further discussion about this.

---

### Official Review · AnonReviewer1 · 2020-03-12
**Automatic segmentation of the pulmonary lobes with a 3D u-net and optimized loss function**

**Rating:** 3
**Confidence:** 4

**Review:**

This paper concerns lobe segmentation from Thoracic CT images using deep learning. This is certainly not the first paper on this topic, but the novelty of this approach is the weighted Dice coefficient, which puts special emphasis on the regions near the lobe boundary. This is not completely novel, as something similar was done already in Gerard et al. Also this paper doesn't have a large amount of data. However, as preliminary work, it is a good idea and shows promise.

---

### Meta-Review · Area_Chair1 · 2020-03-27
**MetaReview of Paper14 by AreaChair1**

**Rating:** 3

**Metareview:**

Reviewers are generally in favor of the methodological contribution of this paper. However, two reviewers complain that there is related work from Gerard et al. in TMI which actually proposes to use fissures for lobe segmentation in a deep learning based framework. The work presented here reproduces the validity of this approach, which is a beneficial finding and thus interesting to report at MIDL 2020. However, these critical reviewer comments are crucial and need to be addressed in a final version of the paper.

**Paper Type:**

validation/application paper

---

### Decision · Program_Chairs · 2020-04-11

Accept